# Urinary Aflatoxin M1 Concentration and Its Determinants in School-Age Children in Southern Ethiopia

**DOI:** 10.3390/nu14132580

**Published:** 2022-06-22

**Authors:** Tafere Gebreegziabher, Melanie Dean, Elilta Elias, Workneh Tsegaye, Barbara J. Stoecker

**Affiliations:** 1Department of Health Sciences, Central Washington University, 400 E University Way, Ellensburg, WA 98926, USA; melanie.dean@cwu.edu; 2School of Nutrition, Food Science and Technology, Hawassa University, Hawassa P.O. Box 5, Ethiopia; eliltaelias5@gmail.com; 3Department of Nutritional Sciences, Oklahoma State University, 421 Human Sciences, Stillwater, OK 74078, USA; workneh.agidew@okstate.edu (W.T.); barbara.stoecker@okstate.edu (B.J.S.)

**Keywords:** aflatoxin, school age children, southern Ethiopia

## Abstract

Aflatoxins are mycotoxins that can contaminate grains, legumes, and oil seeds. These toxic compounds are an especially serious problem in tropical and sub-tropical climates. The objective of this study was to raise awareness of aflatoxin exposure among primary school children in Shebedino woreda, southern Ethiopia, by measuring urinary aflatoxin M1 (AFM1). The study employed a cross-sectional design and systematic random sampling of children from eight schools in the district. The mean ± SD age of the children was 9.0 ± 1.8 years. Most (84.6%) households were food insecure with 17.9% severely food insecure. Urinary AFM1 was detected in more than 93% of the children. The median [IQR] concentration of AFM1/Creat was 480 [203, 1085] pg/mg. Based on a multiple regression analysis: DDS, consumption of haricot bean or milk, source of drinking water, maternal education, and household food insecurity access scale scores were significantly associated with urinary AFM1/Creat. In conclusion, a high prevalence of urinary AFM1 was observed in this study. However, the relation between AFM1 and dietary intake was analyzed based on self-reported dietary data; hence, all of the staple foods as well as animal feeds in the study area should be assessed for aflatoxin contamination.

## 1. Introduction

Aflatoxins (AFs) belong to a group of toxic substances called mycotoxins and are produced by *Aspergillus flavus* and *Aspergillus parasiticus* [1]. They are harmful contaminants that affect a wide variety of food crops including maize, sorghum, groundnuts, wheat, rice, and soy, as well as animal feed, particularly in tropical and subtropical climates [2]. Hot and humid environments and extreme weather create favorable conditions for the *Aspergillus* species to grow and produce toxin. A relative humidity of 80–85% and a temperature around 30 °C are their ideal growth conditions [3]. In addition, high AF exposure has been observed towards the end of the rainy season in hot areas [4,5].

The extent of AF contamination also varies by geographical location, agricultural practice, storage conditions, and the processing of foods [6]. Some studies indicated that exposure to AF is high in rural areas and in low-income women, and the exposure is seasonal [7,8]. The most prevalent and toxic compound, AFB1, contaminates food commodities in tropical and sub-tropical climates. Its metabolite, AFM1, can be assessed in human urine and breast milk, as well as in milk of other species, and is a useful indicator for determining short-term exposure [9]. Moreover, a significant correlation has been observed between dietary AFB1 and urinary AFM1 in humans [10].

Studying the nature, distribution, and sources of AFs is critical because they can cause serious health problems and may also affect the growth and development of young children. AFs are carcinogenic and mutagenic mycotoxins. They have been classified as one of the major causes of cancer and are considered accountable for more than 28% of the cases of liver cancer globally [9,11,12]. In Somalia, because various types of grains have been affected by AF contamination and estimates are that 75 out of 100,000 people die per year from liver cancer due to their consumption of AF-contaminated maize, AF is considered a public health concern [13]. Studies from sub-Saharan Africa showed that AF was linked to poor growth, kwashiorkor, marasmus, micronutrient deficiency, and the impaired immunity of young children [14]. In Uganda, maternal AF exposure during pregnancy was associated with poor birth outcomes such as low birth weight and small head circumference [15].

The occurrence of AF in food stuffs in some parts of Ethiopia has been assessed. As a result, AF contamination was found in various types of staple foods including groundnut, red pepper, maize, malt barley, and sorghum, and AFM1 has been measured in human breast milk and cow’s milk as well as in complementary foods for young children [16,17,18].

However, there is little information in Ethiopia on the prevalence of human exposure to AF, and to the best of our knowledge the current study is the first to report AF exposure in primary school children in Ethiopia. Hence, the objective of this study was to raise awareness of aflatoxin exposure among primary school children in Shebedino woreda, southern Ethiopia, by measuring urinary aflatoxin M1 (AFM1).

## 2. Materials and Methods

### 2.1. Description of Study Area and Population

The study was conducted in Shebedino woreda, Sidama Zone, SNNPR. Shebedino woreda, located in the Great Rift Valley, lies approximately 1710 m above sea level and is a tropical climate. The average temperature is between 18–25 °C. The area is characterized by seasonal and variable rainfall with average annual rainfall between 900–1100 mm [19]. The study population depends on staple foods such as maize, haricot beans, and enset (false banana). Most of the population depend on mixed farming where farmers support their livelihoods from crop production and animal husbandry [19].

### 2.2. Study Design

This cross-sectional school-based study was conducted from May to June 2017 as part of a study examining prevalence of iodine deficiency. Study subjects were randomly selected using a single proportion sample size calculation formula. A sample size of 408 school age children was computed and allocated to eight randomly selected schools by population proportional to size. Students were randomly selected from all children 6–12 years of age in those schools as explained in further detail in our previously published manuscript [20].

### 2.3. Questionnaire

Demographic and socio-economic characteristics of study participants were assessed using questions adapted from the Ethiopian Demographic and Health Survey 2011 report [21]. Food consumption patterns of households and children were estimated using a standardized food frequency questionnaire [22]. Household food insecurity was assessed using the Household Food Security Access Scale (HFIAS) developed by the Food and Nutrition Technical Assistance (FANTA) project of the United States Agency for International Development (USAID) [23]. Children’s dietary diversity score was assessed based on the FAO recommendation [24].

### 2.4. Urinary AFM1 and Creatinine Measurements

Urine samples were collected from each participant in a wet season (May to June). Short term AF exposure was assessed by analyzing urine samples for AFM1 using a commercially available enzyme-linked immunosorbent assay (ELISA) kit for quantitative determination of aflatoxin in urine (Helica Biosystems Inc., Santa Ana, CA, USA). Briefly, the kit contained a microwell plate in which all wells had been coated with an antibody with high affinity for AFM1. When added to a microwell, the AFM1 in the urine sample bound to the coated antibody. Next, a reagent containing aflatoxin bound to horse-radish peroxidase (HRP) was added to each well. After the incubation period, wells were decanted and washed with a PBS buffer with 0.05% Tween20. An HRP substrate was added to each well followed by a stop solution. The microwell plate was then read at 450 nm on a plate reader. Six standards provided with the kit established a standard curve from 0 to 200 pg/mL of aflatoxin. All standards and reagents were provided with the kit. In our hands, the limit of detection was 1.25 pg/mL of aflatoxin and the limit of quantification was 2.5 pg/mL. Urinary creatinine was analyzed using a BioLis 24i clinical chemistry analyzer with standard reagents (Carolina Liquid Chemistries Corp., Brea, CA, USA).

## 3. Statistical Analysis

Frequency distributions, percentages, and means ± SD were used as appropriate in describing the socio-economic and demographic characteristics of the respondents, as well as the children’s urinary AFM1 concentrations. The skewed creatinine-adjusted AFM1 data were log transformed before analysis.

Multiple regression analysis was used to examine the relation between independent variables and AFM1/Creat. The possible collinearity was tested by the calculation of a variance inflation factor (VIF) for all the independent variables; variables that met the criteria (VIF ≤ 4) were entered into the model. All the analyses were performed with SPSS (version 20; IBM. Armonk, NY, USA). Level of significance was set at *p* < 0.05.

## 4. Results

The characteristics of the study participants and parents are presented in Table 1. The mean age of the school children was 9.0 ± 1.8 years and for their mothers was 35.0 ± 7.7 years old. The majority of the children and their mothers were 6 to 9 and 30–39 years old, respectively. Of the children, 20.3% were stunted. The average household size was 5.9 ± 1.7. Over 52% of the mothers were illiterate and 48.5% were involved in farming activities for their living. Except for 63 (15.4%) households, the rest (84.6%) were food insecure with different levels of insecurity. The majority of the households consumed maize, enset, kale, and haricot bean frequently.

More than 93% of the children had AFM1 in their urine above the detection limit of 1.25 pg/mL. The median [IQR] concentration of AFM1/Creat was 480 [203, 1085] pg/mg. The levels of AFM1/Creat varied among the age groups but were not significantly different by age (Figure 1).

In Table 2, the results from a best-fitting multiple regression model with twelve predictor variables for creatinine-adjusted AFM1 concentration are presented. The model included selected socioeconomic variables, of which seven were significant predictors of urinary AFM1/Creat. Maternal education and higher child DDS predicted lower AFM1/Creat and the rest were positive predictors. For every additional increase in the level of maternal education, there was an average reduction of 0.13 pg/mg in urinary AFMI/Creat concentration. For every increase in the number of household members, the AFM1/creat concentration increased by an average of 0.36 pg/mg. On the other hand, a unit increase in the frequency of haricot bean or cow’s milk consumption resulted in 0.087 pg/mg and 0.045 pg/mg increase of AFM1/creat level, respectively. There was an average reduction of AFMI/creat concentration by 0.13 pg/mg for every additional unit increase in DDS. As either the severity of household food insecurity increased or the source of drinking water increased by one unit, the concentration of AFM1/creat increased by 0.18 pg/mg and 0.11 pg/mg, respectively.

## 5. Discussion

The present study assessed whether or not there was aflatoxin in the food supply consumed by Ethiopian primary school children by measuring its metabolite (AFM1) in urine. Further, the potential relations between urinary AFM1 and self-reported food consumption as well as socio-demographic and economic characteristics of households were investigated. AFM1 was detected in the urine of 93% of the 408 children with a high variability in the concentrations. The children’s ages ranged from 6 to 12 years and variable amounts of urinary AFM1 were detected among these age groups. The median [IQR] urinary AFM1/creat was 480 [203, 1085] pg/mg. Our results were extremely high compared to a previous report from Ethiopia, where 17% of children aged 1–4 years showed various types of AFs in urine. Seven percent of these young children had detectable urinary AFM1 with a mean concentration of 64 pg/mL [25].

Our study population was highly food insecure and household food insecurity was a significant predictor of urinary AFM1. In Ethiopia, most farming communities are food insecure, and have a large family size and limited education. All of these factors were associated with higher urinary AFM1 concentration. Although some East African countries (particularly Tanzania and Kenya) have delved deeply into AF research, especially in the last decade, the awareness of AF risk is still in a nascent stage in Ethiopia. One study reported that more than 94% of 360 farmers surveyed had never heard of AF and were not taking steps to mitigate its formation. In half of the households, grains were stored in traditional ‘*gotera*’ made of wood, mud, and straw. Only about 40% of women sorted out moldy grains that were being stored for household use [26].

AF exposure may be different by economic development, living area, and occupation. According to the International Agency for Research on Cancer (IARC), AF exposure is higher in low-income than high-income countries [27]. Studies have shown that the occurrence of AF is significantly higher in Southeast Asia compared to high-income countries from Western Europe [28]. In Taiwan, AF exposure declined following economic development [29], and in Bangladesh, rural inhabitants have shown a high exposure to AF compared to urban inhabitants [30]. This could be explained by the fact that low-income countries and rural communities highly depend on agricultural products and are involved in farming activities in which the contamination of AF in foods and agricultural products is high. In Kenyan school-aged children, a large family size has been associated with high AF exposure [31].

Various studies have reported that high exposure to AF is directly related to the frequent consumption of certain food products such as nuts, nut products, cereals, and spices and herbs [32]. In the present study, children who consumed haricot bean and cow’s milk frequently had higher mean urinary AFM1 than those who consumed these foods less frequently. Other than nuts and cereals, legumes are also suitable for the growth of the fungi that produce AF [33]. Dairy products could be contaminated from animals that consume contaminated feed [18,34]. In Tanzania, a high level of AFB1 was detected in sunflower-based seedcake feed and its metabolite AFM1 was detected in the raw milk from cows fed sunflower seedcakes [35]. In Kenya, among 830 animal feed samples and 613 milk samples most were positive for AFB1 and AFM1, respectively, and the majority of the samples exceeded the FAO/WHO limits [36]. In Turkey, 26.3% of the 76 dairy cow feed samples were contaminated with AFB1, and 21.1% of the milk and milk product samples were contaminated with AFM1 [37]. Similarly, in Japan, corn supplied to cows was contaminated with AFB1 and their milk was contaminated with AFM1 with different levels at different seasons [38]. In Bishoftu town in Ethiopia, AFM1 was detected in samples of milk and milk products collected from industrial and local dairy producers. Researchers commented that this could be a serious health problem for infants and young children due to their relatively high consumption of dairy products [39].

Household sources of drinking water comprised a significant predictor of AFM1 concentration in the current study. Those children who did not have access to a protected water source were more likely to be exposed to AF. In Portugal, different metabolites of AF have been detected in untreated surface water. The related study showed that fungi can produce mycotoxins in water which could cause detrimental health effects [40]. Even a low dose presence of mycotoxins in treated drinking water has raised a concern for future public health in South Africa [41]. Moreover, AFB1, AFB2, AFG1, and ochratoxin A were detected in different commercially available brands of bottled water in Portugal [42].

These fungi are very common in the tropical and sub-tropical geographical areas where the weather is mostly warm and humid, which creates suitable conditions for their growth. In Malaysia, a high amount of AFB1 was detected in nuts and nut products, and among the major contributing factors were the warm and humid weather conditions [43]. Moreover, the improper storage of foods and agricultural products promotes the growth of the fungi, and subsequently the contamination of these products by AF [32,44]. In Indonesia, the tropical climate and inappropriate food safety practices contributed to the AFB1 contamination of maize and nuts [45]. Seasonal variation and post-harvest practices also contribute to an increased AF contamination [7,46].

High levels of AFB1 were found in maize in Ethiopia in some villages nearby the study area [26,47]. In the present study, the consumption of maize was not significantly associated with the concentration of urinary AFM1. This does not indicate that the maize was free of contamination; however, maize is the staple food, and everybody in the area concerned eats food made from maize at least once a day. This lack of variability might have masked any potential underlying associations. In Sub-Saharan Africa, young children are highly exposed to AF through contaminated foods such as maize and peanuts [4,48]. In Tanzania, contaminated maize consumption was a main cause of AF exposure in children aged 24–36 months associated with breastfeeding and weaning foods [49]. In Ethiopia, maize and raw cow’s milk from different places and agro-ecological zones were contaminated by high levels of AF and confirmed as a serious threat to smallholder farmers [47,50]. However, the investigation of AF contamination of crops in Ethiopia is poorly regulated. Due to the limited resources, very few food products are effectively tested for AF contamination, especially in the context of local markets [51,52]. Although efforts towards minimizing AF contamination in the peanut production system are being attempted, it was commented that the intercropping of peanuts with maize and/or sorghum could create favorable conditions for the distribution of AFs [53]. One study reported AF contamination in 29% of maize samples collected from eastern, central, and western regions of Ethiopia and found that the mean concentration of AF was 53 ppb, which is well above the European Union (EU) and the Food and Drug Administration (FDA) action levels for AF the contamination of products meant for human consumption [47]. A lack of regulation enforcement at the local level combined with climatic and storage conditions conducive to high levels of AF contamination creates a public health concern especially in communities of subsistence farmers, such as those in Ethiopia. Kenya has experienced as high as 8000 µg/kg of AF in maize, and AF was the cause of 317 case-patients and the death of 125 people in 2004 [54]. In the Eastern Democratic Republic of Congo, AF was found in all of the 215 maize samples tested with the levels ranging from 0.3 to 18.5 µg/kg [55].

Although most subsistence farmers have limited dietary diversity, in the present study, access to more diversified food was associated with lower exposure to AFM1 and a more diversified diet was a significant predictor of a decreased AFM1 concentration. Some studies suggested that access to improved dietary diversity could lower the risk of exposure to AF by minimizing the amount of toxin that could be consumed from monotonous contaminated food commodities [56,57]. In Sub-Saharan Africa and parts of Asia, some commonly consumed crops such as maize, groundnuts, and cassava are important sources for foodborne toxins. Relying on one or two types of food would increase the susceptibility to an excessive intake of certain toxins which could lead to various types of health complications such as cancer, neurological disorders, and immunotoxicity, as well as growth impairment [58]. In Uganda, a limited diet diversity aggravated the risk of exposure to AF [59]. Apart from that, it is evidenced that dietary diversity provides nutrient adequacy and improves the nutritional status of children [60].

This study has several limitations. The Data were collected in a short period of time; hence, the results do not show the possible seasonal variation of exposure to AF. In addition, urinary AFM1 reflects exposure to AF in the past few days; however, in subsistence households where families consume their own crops, AF contamination of stored grain may cause long periods of exposure to the toxin. This research was not designed to track the measured urinary AFM1 back to its specific source in the food supply; nonetheless, the results can raise awareness of a public health problem that has received too little attention to date in Ethiopia.

## 6. Conclusions

In conclusion, the results from this study showed a high concentration of urinary AFM1 in school aged children from Shebedino woreda, southern Ethiopia. AFM1 was associated with maternal educational level, household food insecurity status, and the source of drinking water. However, the relation between AFM1 and food consumption was analyzed based on self-reported data; hence, it is recommended that all of the staple foods and animal feeds in the study area should be assessed for AF contamination in order to take preventive measures. Moreover, because the toxin causes detrimental health problems, an assessment of the short- and long-term exposure in relation to health status is warranted.

## Figures and Tables

**Figure 1 nutrients-14-02580-f001:**
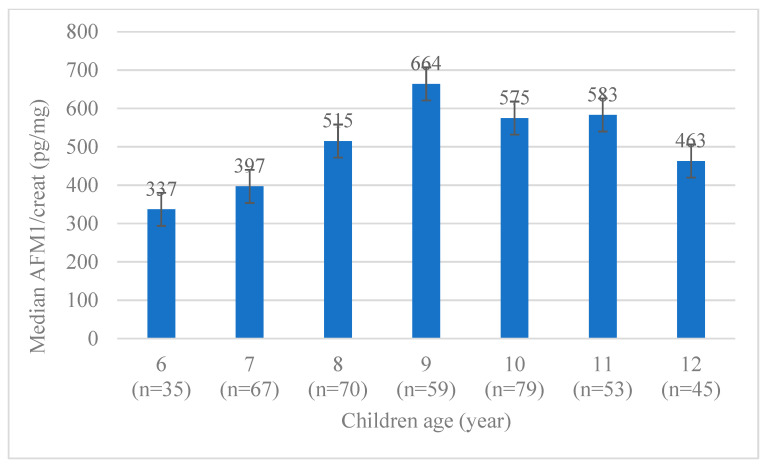
Urinary AFM1/Creat concentration of school children by age (*n* = 408).

**Table 1 nutrients-14-02580-t001:** Socio economic characteristics and food consumption patterns of respondents and children (*n* = 408).

Variables	Frequency (%)	Mean (SD)
Child age (years)		9 (1.8)
- 6–9	231 (56.6)	
- 10–12	177 (43.4)	
Child sex		
- Male	202 (49.5)	
- Female	206 (50.5)	
HAZ score ^a^		
- Stunted	83 (20.3)	
- Not stunted	325 (79.7)	
Mother’s age (years)		35.0 (7.7)
- 21–29	78 (19.1)	
- 30–39	229 (56.1)	
- 40–49	77 (18.9)	
- ≥50	24 (5.9)	
Household size		5.9 (1.7)
- 2–4	177 (43.4)	
- 5–7	203 (49.8)	
- 8–11	28 (6.8)	
Mother’s education		
- Illiterate	214 (52.5)	
- Literate	194 (47.5)	
Mother’s occupation		
- Farming	198 (48.5)	
- Non-farming	210 (51.5)	
HFIAS ^b^		
- Food secure	63 (15.4)	
- Mild food insecurity	123 (30.2)	
- Moderate food insecurity	149 (36.5)	
- Severe food insecurity	73 (17.9)	
Frequency of maize consumption		
- Once or more per day	387 (94.9)	
- Sometimes ^c^	14 (3.4)	
- Rarely/Never ^d^	7 (1.7)	
Frequency of haricot bean consumption		
- Once or more per day	107 (26.2)	
- Sometimes	95 (23.2)	
- Rarely/Never	206 (50.6)	
Frequency of milk consumption		
- Once or more per day	14 (3.4)	
- Sometimes	17 (4.2)	
- Rarely/Never	377 (92.4)	
Frequency of enset consumption		
- Once or more per day	352 (86.3)	
- Sometimes	46 (11.3)	
- Rarely/Never	10 (2.4)	
Frequency of kale consumption		
- Once or more per day	202 (49.5)	
- Sometimes	184 (45.1)	
- Rarely/Never	22 (5.4)	

^a^ HAZ score—Height for age Z score, ^b^ HFIAS—Household food insecurity access scale, ^c^ Sometimes—2–6 times a week, ^d^ Rarely/Never—0–4 times per month.

**Table 2 nutrients-14-02580-t002:** Multiple regression predicting urinary AFM1/creatinine (pg/mg) concentration of school age children (*n* = 402).

Variable	β	95% (CI)	*p*
Constant	1.92	1.31, 2.56	
Children age	−0.021	−0.05, 0.036	0.36
Maternal education	−0.13	−0.19, −0.011	**0.025 ***
Father’s education	0.034	−0.05, 0.11	0.56
Household size	0.36	0.10, 0.64	**0.020 ***
HFIAS ^a^	0.18	0.058, 0.24	**0.001 *****
DDS ^b^	−0.13	−0.25, −0.035	**0.004 ****
Frequency of wheat consumption	0.074	−0.023, −0.149	0.148
Frequency of haricot bean consumption	0.087	0.012, 0.171	**0.025 ***
Frequency of cow milk consumption	0.045	0.008, 0.084	**0.026 ***
Frequency of maize consumption	−0.01	−0.096, 0.08	0.45
Frequency of enset consumption	−0.05	−0.098, 0.07	0.52
Source of drinking water ^c^	0.11	0.064, 0.495	**0.01 ****

R^2^ = 13.5, *** Significant α.001, ** Significant α.01, * Significant α.05, ^a^ Household food insecurity access scale, ^b^ DDS—Dietary diversity score, ^c^ Source of drinking water was coded as 1—public tap, 2—protected well/spring, 3—unprotected well/spring.

## Data Availability

The dataset used and/or analyzed during the current study are available from the corresponding author on reasonable request.

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
