# Peer review of "Urinary Aflatoxin M1 Concentration and Its Determinants in School-Age Children in Southern Ethiopia"

_nutrients, 2022, doi:10.3390/nu14132580_

Round 1

Reviewer 1 Report

Dear, considering submitted manuscript for review my opinion are as follows.

In the investigated area the authors tried to find link among Demographic and socio-economic factors and AFM1 concentration in urine of children. The theme of presented manuscript is actual, primarily because describes factors affecting mycotoxins exposure.

However, the authors did not provide enough data to make the work systematic or interesting, and the discussion is also poor. The research protocol used was not adjusted to the research hypothesis, making the research protocol completely non-physiological and unacceptable. In section Materials and Methods, sampling and samples are poor described. Is is unknown how many children’s and samples of urines are analysed per investigated group. Results are not clear in some case. E.g. in figure and tables I and therefore is hard to read and understand. Distribution of AFM1 concentration among investigated groups are not well presented. Considering that number of samples is missing; results in Fig 1 can’t be taken as relevant. Characteristics of study participants, particularly mothers is not explained in section results and discussed, what is important for the study. How can you explain link/statistical significance between mother’s characteristics, Household size, and other variables and AFM1 concentration in urine, shown in table 3? Data regarding to food consumption paterns are missing and therefore link between food consumption and demographic and socio-economic factors are not well conducted and consequently well explained in section Discussion.

In section Results and Discussion a lot of repetition of statement was found and the discussion is also poor. In discussion authors' obtained results should be explained for proven their findings and compared with others references, rather than just listed references. It should be more specified, what others cited authors had find and their results in similar context. Therefore, the discussion should be more extensive.

In current form manuscript does not meet scientific and journal criteria for publishing.

Author Response

Reviewer 1.

In the investigated area the authors tried to find link among Demographic and socio-economic factors and AFM1 concentration in urine of children. The theme of presented manuscript is actual, primarily because describes factors affecting mycotoxins exposure.

However, the authors did not provide enough data to make the work systematic or interesting, and the discussion is also poor. 

  • We have added Food Consumption Frequency to Table 1. The discussion has improved.

 The research protocol used was not adjusted to the research hypothesis, making the research protocol completely non-physiological and unacceptable.

  • We have attempted to rectify this throughout the manuscript.

 In section Materials and Methods, sampling and samples are poor described. Is is unknown how many children’s and samples of urines are analysed per investigated group.

  • Sampling technique was explained in our previously published manuscript and reference was provided. Below is what we had in the published manuscript “Sample size A single proportion sample size calculation formula was used to determine optimal number for estimating the prevalence of iodine deficiency. A sample size of 408 school age children was computed based on an estimated 59.1% prevalence of goiter [9], a 95% confidence level, 5% margin of error, design effect of 1.5 and nonresponse rate of 10%. Sampling technique The woreda has 37 primary schools, out of which eight schools were selected by simple random sampling. The sample was allocated for each school by population proportional to size. The sampling frame was prepared by using registers that listed children 6–12 years of age in all selected schools. Finally, systematic random sampling was used to select the study participants from each school. Sample size is included in Line 80 - 81 and a brief description of the sampling technique is now added to Line 81 - 83.

 Results are not clear in some case. E.g. in figure and tables I and therefore is hard to read and understand. 

  • We have clarified data in Figure 1 and Table 1.

Distribution of AFM1 concentration among investigated groups are not well presented. Considering that number of samples is missing; results in Fig 1 can’t be taken as relevant.

  • Number of children in each age group have been added to Figure 1.

Characteristics of study participants, particularly mothers is not explained in section results and discussed, what is important for the study. How can you explain link/statistical significance between mother’s characteristics, Household size, and other variables and AFM1 concentration in urine, shown in table 3?

  • .The following sentence has been added in the discussion section. ‘Culturally, mothers are responsible for securing and preparing household food and their level of education may contribute to knowledge about the risk of exposure to AF’ (please see line number 173 – 174).

 Data regarding to food consumption paterns are missing and therefore link between food consumption and demographic and socio-economic factors are not well conducted and consequently well explained in section Discussion.

  • Information regarding frequency of consumption of common food items has been added to Table 1 and explained in the results section (please see line numbers 117 – 118).

In section Results and Discussion a lot of repetition of statement was found and the discussion is also poor. 

To avoid the repetition mentioned, we have decided to remove the multiple classification analysis results (Table 2) and concentrate on a clear presentation of the results from the multiple linear regression model. 

In discussion authors' obtained results should be explained for proven their findings and compared with others references, rather than just listed references. It should be more specified, what others cited authors had find and their results in similar context. Therefore, the discussion should be more extensive.

  • The following results of cited references have been added to the discussion as suggested. ‘In Kenya, among 830 animal feed samples and 613 milk samples, most were positive for AFB1 and AFM1 respectively, and the majority of the samples exceeded the FAO/WHO limits [33]. In Turkey 26.3 % of the 76 dairy cow feed samples were contaminated with AFB1 and 21.1% of the milk and milk product samples were contaminated with AFM1[34]. Similarly in Japan, corn supplied to cows was contaminated with AFB1 and their milk was contaminated with AFM1 with different levels at different seasons (Please see line number 200 – 205). However, investigation of AF contamination of crops in Ethiopia is poorly regulated.  Because of limited resources, very few food products are effectively tested for AF contamination, especially in the context of local markets [49, 50] One study found AF contamination in 29% of maize samples collected from east, central, and west regions of Ethiopia and found that the mean concentration of AF was 53 ppb which is well above the European Union (EU) and the Food and Drug Administration (FDA) action levels for AF contamination of products meant for human consumption [44].  A lack of regulation enforcement at the local level, combined with climatic and storage conditions conducive to high levels of AF contamination creates a public health concern especially in communities of subsistence farmers, such as those in Ethiopia.   (please see line numbers 235 – 244)’.

Reviewer 2 Report

May 30th, 2022

Letter to Authors

 I have read and reviewed your manuscript titled Aflatoxin exposure and its determinants in school-age children in southern Ethiopia” (nutrients-1755423) submitted to NUTRIENTS.

Manuscript presents valuable results about AF exposure in school-age children, using Elisa tests. After reading of your manuscript I could recommend it to be published after attending the next MINOR REVISIONS. In order to improve the quality of the manuscript I suggest a few corrections:

Comments:

1)      Introduction. No comments.

2)      Material and Methods. It should be included next information in the section:

1) more information about number of children involved in the studies, although in table 3 are number n=402, in other tables ‘n’ seems to be different.

2) more details of Elisa tests like the preparation of buffers,

3) the method of extraction of the mycotoxin,

3)      Results

·         In table 1, some values are given in format ‘.164’ and some in format with ‘0’ (i.e. ‘0.138’). It should be corrected.

·         All abbreviations used in tables should be clearly explained in the legend (i.e. HAZ, HFIAS).

·         On figure 1, the standard deviations ranges at the bars are missing and it is would be worth adding the number of each age group, because it is interesting aspects of studies and in table 2 there are only the number for two age group (6-9 and 10-12 years).

·         Please, correct font size in heading of Table 3.

4)      Discussion. The discussion is sufficient and rich with many examples not only from Africa, but also from Asia and Europe.

5)      Conclusions. They should be included in separate chapter. The main results and influencing factors should be reported in a few sentences.

I really thank you for your consideration, and I sincerely hope these recommendations could be useful to you to improve the quality of your manuscript since the item might be important to potential readers of Nutrients.

Author Response

Reviewer 2.

 I have read and reviewed your manuscript titled Aflatoxin exposure and its determinants in school-age children in southern Ethiopia” (nutrients-1755423) submitted to NUTRIENTS.

Manuscript presents valuable results about AF exposure in school-age children, using Elisa tests. After reading of your manuscript I could recommend it to be published after attending the next MINOR REVISIONSIn order to improve the quality of the manuscript I suggest a few corrections:

Comments:

1)      Introduction. No comments.

2)      Material and Methods. It should be included next information in the section:

1) more information about number of children involved in the studies, although in table 3 are number n=402, in other tables ‘n’ seems to be different.

  • Number of children involved in the study now is included under study design and in tables.

2) more details of Elisa tests like the preparation of buffers,  

  • The ELISA tests were commercially available kits and all buffers were prepared according to the manufacturer’s instructions (Please see line number 94 – 96).

 3) the method of extraction of the mycotoxin.

  • The AFM1 was not extracted from the urine samples.  Instead each diluted urine sample and standard was reacted with the AFM1-specific antibody bound to the walls of the microwell plate in the commercial direct enzyme-linked immunosorbent assay that we used.   

3)      Results

  • In table 1, some values are given in format ‘.164’ and some in format with ‘0’ (i.e. ‘0.138’). It should be corrected.
  • Values corrected as suggested.
  • All abbreviations used in tables should be clearly explained in the legend (i.e. HAZ, HFIAS).
  • Abbreviations are explained in the legend as suggested.
  • On figure 1, the standard deviations ranges at the bars are missing and it is would be worth adding the number of each age group, because it is interesting aspects of studies and in table 2 there are only the number for two age group (6-9 and 10-12 years).

  • Error bars and number of subjects in each age group have been included in figure 1.
  • Please, correct font size in heading of Table 3.

- Font size in heading of tables  has been corrected.

4)      Discussion. The discussion is sufficient and rich with many examples not only from Africa, but also from Asia and Europe. Thank you.

5)      Conclusions. They should be included in separate chapter. The main results and influencing factors should be reported in a few sentences.

  • Conclusion is now in a separate section. The main results are reported in the conclusion section (please see line number 261 – 263).

I really thank you for your consideration, and I sincerely hope these recommendations could be useful to you to improve the quality of your manuscript since the item might be important to potential readers of Nutrients.

Reviewer 3 Report

The study described here is of interest and well conducted and described. Some additional work in terms of results explanation and impact is needed.

1. the authors should try and identify the responsible food element for the levels found in children . Check in literature and see if eg milk (eg ref Food Chem Toxicol. 2020 Aug;142:111455. doi: 10.1016), or nuts or any other food stuff is the main contributor.

2. Do you have any measurements available of aflatoxins or mycotoxins (eg ref Toxicol Rep. 2021 Nov 5;8:1856-1864. doi: 10.1016) in common food stuff from your country and the respective risk assessment?

3. Could you correlate or at least discuss the health status of the children , preferably your study population, with regards to the levels of exposure?

4. Could you provide information on the legal, monitoring and control system followed in your country regarding aflatoxins and food contaminants in general? Number of controls and non-compliances would be interested and helpful to correlate them statistically with your findings.

Author Response

Reviewer 3.

The study described here is of interest and well conducted and described. Some additional work in terms of results explanation and impact is needed.

  1. the authors should try and identify the responsible food element for the levels found in children . Check in literature and see if eg milk (eg ref Food Chem Toxicol. 2020 Aug;142:111455. doi: 10.1016), or nuts or any other food stuff is the main contributor.

- The study population depends on very limited types of foods including maize, haricot bean and enset (false banana). For instance, none of the study participants has ever eaten nuts and very few often consumed milk. We have tried to see if there is any association between AFM1 and frequently consumed food stuffs in the study area (please see table 2). The following statement has been included in the methods section. ‘Children’s dietary diversity score was assessed based on the FAO recommendation. (Please see line number 91).

  1. Do you have any measurements available of aflatoxins or mycotoxins (eg ref Toxicol Rep. 2021 Nov 5;8:1856-1864. doi: 10.1016) in common food stuff from your country and the respective risk assessment?

- References that reported aflatoxin contamination in different foods in Ethiopia have been cited in our study (please see reference numbers 16, 17, 30, 36 and 49).

  1. Could you correlate or at least discuss the health status of the children, preferably your study population, with regards to the levels of exposure?

- We haven’t assessed health status of the children in this study, but studies in other countries have suggested associations between aflatoxin exposure and poor health outcomes.  Please see line 245 - 246 and 253 – 256 for some examples.

  1. Could you provide information on the legal, monitoring and control system followed in your country regarding aflatoxins and food contaminants in general? Number of controls and non-compliances would be interested and helpful to correlate them statistically with your findings.

- The following statement has been included in the discussion section.

AF contamination of crops in Ethiopia is poorly regulated.  Because of limited resources, very few food products are effectively tested for AF contamination, especially in the context of local markets (Please see line number 236 – 245).

Round 2

Reviewer 1 Report

Dear, considering the manuscript submitted after the first round of revision, my opinion is that despite the effort manuscript still does not meet the scientific criteria for publishing. 

Research has been conducted in a very limited period (two months). Therefore, obtained results, and statistical significance has some limitations and weaknesses which should be taken into the consideration during the discussion. 

In addition, one of the main shortcomings of the study is the lack of traceback to estimate ingested AF on the basis of AFM1 concentration in urine. Also, the rate of concern was not evaluated and taken into consideration. Preventive measures not proposed and future directions on research.

A major revision is requested. 

Author Response

Dear, considering the manuscript submitted after the first round of revision, my opinion is that despite the effort manuscript still does not meet the scientific criteria for publishing. 

Did you consider the impact of the hepatitis virus?

Development of hepatocellular carcinoma (HCC) is related to chronic aflatoxin intake.  Because HCC  may take up to 20 years to present, it is difficult to tie it to specific aflatoxin exposure.  There is considerable concern simultaneous infection with the hepatitis B virus and exposure to aflatoxin may cause more rapid development of HCC but because participants were only primary school students there was not likely to be a demonstrable relation between the hepatitis B and aflatoxin exposure in terms of health in these young children.  Furthermore, this was a community-based study, not a clinical study, and testing for hepatitis B was beyond the scope of our study. 

An excellent review has recently been published.   Meijer, N., Kleter, G., de Nijs, M., Rau, M.-L., Derkx, R., & van der Fels-Klerx, H. J. (2021). The aflatoxin situation in Africa: Systematic literature review. Comprehensive reviews in food science and food safety, 20(3), 2286-2304. doi:10.1111/1541-4337.12731

Describe the used methodology in brief. Also quality assurance of used analytical method, parameters of validation (LOD, LOQ etc).. 

We have included the following points to the method section (please see line number 100 – 114)

AFM1 concentrations were determined using a commercially available direct enzyme-linked immunosorbent assay (ELISA) kit and manufacturer’s instructions were followed meticulously.  The urine sample was initially diluted with distilled water and then mixed with assay buffer (provided) and added to a microwell plate in which all wells had been coated with an antibody with high affinity for AFM1.  AFM1 in the urine sample bound to the coated antibody.  Next aflatoxin bound to horse-radish peroxidase (HRP) (provided) was added to each well and incubated in the dark at room temperature.  The aflatoxin conjugate bound to all antibody not already occupied by aflatoxin from the sample.  After the incubation period, wells were decanted and washed.  An HRP substrate (provided) was added to each well.  This substrate developed a blue color that was inversely proportional to the amount of aflatoxin in the sample.  An acidic stop solution was added to each well which changed the chromogen from blue to yellow.  Then the microwell plate was read at 450 nm.  Six standards provided with the kit established a standard curve from 0 to 200 pg/mL of aflatoxin.  This set of standards was analyzed with each microwell plate.  The lowest standard after 0.0 was 2.5 pg/mL.  In our hands, the LOD was 1.25 pg/mL of aflatoxin and the LOQ was 2.5 pg/mL. Urine samples with aflatoxin concentrations above 200 pg/mL were diluted further and re-analyzed.  

Research has been conducted in a very limited period (two months). Therefore, obtained results, and statistical significance has some limitations and weaknesses which should be taken into the consideration during the discussion. 

We have included a paragraph citing some of the limitations of the study. Please see Line number 288 - 294.

In the discussion section could you explain point by point a link between AFM1/Creat and variables in table 2. Particularly the statistical significant.

Significant variables have been explained point by point in the discussion section.

In addition, one of the main shortcomings of the study is the lack of traceback to estimate ingested AF on the basis of AFM1 concentration in urine. Also, the rate of concern was not evaluated and taken into consideration. Preventive measures not proposed and future directions on research.

The following paragraph has been included in the discussion section

Although some East African countries (particularly Tanzania and Kenya) have delved deeply into aflatoxin research especially in the last decade, awareness of aflatoxin risk is still in a nascent stage in Ethiopia.  One study reported that more than 94% of 360 farmers surveyed had never heard of it and were not taking steps to mitigate aflatoxin formation (Boshe et al. 2020).  In half of the households, grains were stored in traditional gotera made of wood mud and straw.  Only about 40% of women sorted out moldy grains that were being stored for household use.  Clearly there is an important awareness gap. (Please see line number 200 – 205). 

We acknowledge that urinary AF is a measure of recent exposure, we didn’t attempt to distinguish the source, we were trying to raise awareness. The lack of awareness of aflatoxin as a problem needs to be addressed at the level of the farmer, the subsistence household, the marketing organizations accepting grain, and the consumer.  

 The following statements have been included as preventive measures and future research directions.

the relation between AFM1 and the independent variables was analyzed based on self-reported data, hence it is recommended that all of the staple foods as well as animal feeds in the study area should be assessed for AF contamination in order to take preventive measures. Moreover, because the toxin causes detrimental health problems, assessment of short and long term exposure in relation to health status is warranted (Please see line number 299 – 303).

A major revision is requested.